# One-Year Changes in Urinary Microbial Phenolic Metabolites and the Risk of Type 2 Diabetes—A Case-Control Study

**DOI:** 10.3390/antiox11081540

**Published:** 2022-08-08

**Authors:** María Marhuenda-Muñoz, Inés Domínguez-López, Emily P. Laveriano-Santos, Isabella Parilli-Moser, Cristina Razquin, Miguel Ruiz-Canela, Francisco Javier Basterra-Gortari, Dolores Corella, Jordi Salas-Salvadó, Montserrat Fitó, José Lapetra, Fernando Arós, Miquel Fiol, Lluis Serra-Majem, Xavier Pintó, Enrique Gómez-Gracia, Emilio Ros, Ramon Estruch, Rosa M. Lamuela-Raventós

**Affiliations:** 1Centro de Investigación Biomédica en Red Fisiopatología de la Obesidad y la Nutrición (CIBEROBN), Instituto de Salud Carlos III, 28029 Madrid, Spain; 2Department of Nutrition, Food Science and Gastronomy, School of Pharmacy and Food Sciences and XIA, University of Barcelona, 08028 Barcelona, Spain; 3Institute of Nutrition and Food Safety (INSA-UB), University of Barcelona, 08921 Santa Coloma de Gramenet, Spain; 4Department of Preventive Medicine and Public Health, IdiSNA, Navarra Institute for Health Research, University of Navarra, 31008 Pamplona, Spain; 5Department of Endocrinology and Nutrition, IdiSNA, Navarra Institute for Health Research, Hospital Universitario de Navarra, 31008 Pamplona, Spain; 6Department of Preventive Medicine, University of Valencia, 46010 Valencia, Spain; 7Departament de Bioquímica i Biotecnologia, Unitat de Nutrició Humana, Universitat Rovira i Virgili, 43204 Reus, Spain; 8Institut d’Investigació Sanitària Pere Virgili (IISPV), 43201 Reus, Spain; 9Unit of Cardiovascular Risk and Nutrition, Institut Hospital del Mar de Investigaciones Médicas (IMIM), 08003 Barcelona, Spain; 10Research Unit, Department of Family Medicine, Distrito Sanitario Atención Primaria Sevilla, 41013 Sevilla, Spain; 11Department of Cardiology, Hospital Txangorritxu, 01009 Vitoria, Spain; 12Institut Universitari d’Investigació en Ciències de la Salut (IUNICS), 07122 Palma de Mallorca, Spain; 13Department Clinical Sciences, University of Las Palmas de Gran Canaria, 35016 Palmas de Gran Canaria, Spain; 14Lipid Unit, Department of Internal Medicine, IDIBELL-Hospital Universitari de Bellvitge, L’Hospitalet de Llobregat, FIPEC, 08908 Barcelona, Spain; 15Department of Epidemiology, School of Medicine, University of Malaga, 29010 Málaga, Spain; 16Lipid Clinic, Endocrinology and Nutrition Service, Institut d’Investigacions Biomèdiques August Pi Sunyer (IDIBAPS), Hospital Clínic, 08036 Barcelona, Spain; 17Internal Medicine Department, Institut d’Investigacions Biomèdiques August Pi Sunyer (IDIBAPS), Hospital Clinic, University of Barcelona, 08036 Barcelona, Spain

**Keywords:** bioactive compounds, phytochemicals, Mediterranean diet, PREDIMED study, urinary microbial phenolic metabolites, cardiovascular, liquid chromatography, mass spectrometry

## Abstract

The intake of polyphenols has been associated with a risk reduction of type 2 diabetes. Nevertheless, to the best of our knowledge, the molecules that might be metabolically active after ingestion are only starting to be investigated regarding this metabolic disease. To investigate the association between one-year changes in urinary microbial phenolic metabolites (MPM) and the incidence of type 2 diabetes, we performed a case-control study using data and samples of the PREDIMED trial including 46 incident type 2 diabetes cases of 172 randomly selected participants. Eight urinary MPMs were quantified in urine by liquid chromatography coupled to mass spectrometry and used to assess their associations with type 2 diabetes risk by multivariable logistic regression models. Compared to participants in the lowest tertile of one-year changes in hydroxybenzoic acid glucuronide, those in the highest tertile had a significantly lowered probability of developing type 2 diabetes (OR [95% CI], 0.39 [0.23–0.64]; *p* < 0.001 for trend). However, when additionally adjusting for fasting plasma glucose, the statistical significance was lost. Changes in the dietary pattern can increase the concentrations of this compound, derived from many (poly)phenol-rich foods, and might be changing the gut microbial population as well, promoting the production of the metabolite.

## 1. Introduction

An association of the Mediterranean diet (MeDiet) with a reduced incidence of type 2 diabetes was demonstrated in the PREDIMED (PREvención con DIeta MEDiterránea) nutrition intervention trial, conducted in older individuals at high cardiovascular risk [1]. Compared to the control diet group, there was a significant 40% risk reduction by the MeDiet supplemented with extra-virgin olive oil, and a non-significant 18% risk reduction by the MeDiet supplemented with mixed nuts [1]. Subsequently, several analyses were performed to assess the effect of individual components of the MeDiets on this outcome. Thus, Tresserra-Rimbau et al. reported that a higher intake of flavonoids, stilbenes, and total (poly)phenols from the MeDiet was associated with reduced incidence of type 2 diabetes [2].

(Poly)phenols are one of the main group of antioxidants and anti-inflammatory compounds in the MeDiet, and several health-promoting and disease-preventing effects have been attributed to them in both experimental and clinical studies [3,4,5]. After the ingestion of (poly)phenols, their bioavailability is enhanced by modification by the gut microbiota, giving rise to new compounds known as microbial phenolic metabolites (MPMs). These compounds reach the bloodstream and can have a biological effect on target organs. MPMs are transformed by the cytochromes into phase II metabolites (glucuronides and sulphates), which are the active molecules in human fluids [6].

Despite being highly bioactive molecules, given that the number of molecules pertaining to the (poly)phenolic family is large, their quantitation and association with type 2 diabetes onset have not been extensively studied up to date. This pilot study aims to assess the association of urinary-excreted MPM concentrations with type 2 diabetes development risk in PREDIMED participants.

## 2. Materials and Methods

### 2.1. Study Design

This work was performed as a pilot case-control study nested within the PREDIMED trial, a multicenter, parallel-group, randomized, controlled trial that evaluated the effect of the MeDiet on the primary prevention of cardiovascular disease (CVD) on women and men ranging in age from 55 to 80 years at high cardiovascular risk but free of CVD at baseline. Participants (*n* = 7447) were randomized into three intervention groups: Extra-virgin olive oil-supplemented MeDiet, mixed nuts-supplemented MeDiet, and low-fat control diet. Details on the study protocol can be found in Martínez-González et al., 2012 [7] and at http://www.predimed.es (accessed on 16 March 2022). One of the secondary outcomes of the study was incidence of type 2 diabetes; out of 3541 participants free of diabetes at baseline, 273 developed the disease during 4.1 years of follow-up [1]. For the present case-control study, we included a random selection of 172 participants from all eligible subjects of the PREDIMED study who were free of diabetes at baseline and with available urine samples at baseline and after one year of follow-up, of whom 46 were incident cases of type 2 diabetes and 126 were non-cases (Figure 1).

Participants who reported implausible energy intakes (<500 or >3500 Kcal/day for women and <800 or >4000 Kcal/day for men) were eliminated from the selection [8].

The PREDIMED study was conducted according to the ethical standards’ guidelines of the Declaration of Helsinki and the CONSORT guidelines (Appendix A), and all procedures were approved by the Institutional Review Boards of the participating centers. The clinical trial was registered in the ISRCTN with the number 35739639 (https://www.isrctn.com/ISRCTN35739639, accessed on 27 April 2022). Written informed consent was obtained from all participants.

### 2.2. Type 2 Diabetes Case Ascertainment

Information on type 2 diabetes development was registered on a yearly basis and compiled from the continuous contact with participants and their primary care physicians during the trial. New onset of type 2 diabetes was diagnosed according to the American Diabetes Association criteria (>7.0 mmol/L fasting plasma glucose or >11.1 mmol/L 2-h plasma glucose after a 75 g oral glucose overload) [9]. The endpoint was determined by the PREDIMED Clinical Events Committee, whose members were blinded to intervention group. Only cases confirmed by this committee were included in the analyses.

### 2.3. MPM Analysis

The concentration of the different MPMs in urine was calculated after solid phase extraction and liquid chromatography coupled to a high-resolution mass spectrometer, and it was normalized by the creatinine concentration of the urine.

#### 2.3.1. Samples, Standards, Solvents, and Equipment

Fasting morning spot urine samples from the baseline and one-year visits were stored at −80 °C until the analyses. Handling of all samples and standards was always done under filtered light and under cool conditions to prevent phenolic oxidation.

MPM standards were purchased from the following commercial suppliers: 3-hydroxytyrosol, protocatechuic acid, 4-hydroxybenzoic acid, 3,4-dihydroxyphenylpropionic acid, 3′-hydroxyphenylacetic acid, o-coumaric acid, m-coumaric acid, *p*-coumaric acid, enterodiol, urolithin-A, and urolithin-B from Sigma-Aldrich (St. Louis, MO, USA); 3′-hydroxytyrosol-3′-glucuronide, dihydroresveratrol, and (+)*cis,trans*-abscisic acid d_6_ from Santa Cruz (Santa Cruz Biotechnology, Santa Cruz, CA). 3-hydroxybenzoic acid, vanillic acid, syringic acid, and enterolactone from Fluka (St. Louis, MO, USA). Standards were stored in powder form and protected from light.

Creatinine was also purchased from Fluka (St. Louis, MO, USA). Methanol of LC-MS grade and acetonitrile (MeCN) of HPLC grade were obtained from Sigma-Aldrich and formic acid (≥98%) from Panreac Química S.A. (Barcelona, Spain). Ultrapure water (Milli-Q) was generated by a Millipore system (Bedford, MA, USA). Synthetic urine was composed of calcium chloride (0.65 g/L), magnesium chloride (0.65 g/L), sodium chloride (4.6 g/L), sodium sulfate (2.3 g/L), sodium citrate (0.65 g/L), dihydrogen phosphate (2.8 g/L), potassium chloride (1.6 g/L), ammonium chloride (1.0 g/L), urea (25 g/L), and creatinine (1.1 g/L), all diluted in ultrapure water [10].

The solid phase extraction was performed with Waters Oasis HLB 96-well plates 30 µm (30 mg) (Milford, MA, USA). The sample concentrator was obtained from Techne (Duxford, Cambridge, UK). The filters were 0.22 µm polytetrafluoroethylene 96-well plate filters from Millipore (Burlington, MA, USA). For the chromatographic separation we used an Accela chromatograph from Thermo Scientific (Hemel Hempstead, UK) equipped with a quaternary pump and a thermostated autosampler set at 4 °C, all operated by Chromeleon Xpress software 7.3.1 (Thermo Fisher Scientific, San Jose, CA, USA).

#### 2.3.2. MPM Extraction and Analysis

MPMs were extracted and analyzed following a validated method [11]. Briefly, 50 μL of urine were diluted with ultrapure water bringing them to 1 mL, acidified with 2 μL of formic acid, and centrifuged at 15,000× *g* at 4 °C for 4 min. The samples were then subjected to solid phase extraction using reverse phase sorbent 96-well plates that had been activated with 1 mL of methanol and 1 mL of 1.5 M formic acid, added consecutively. 1 mL of sample was loaded into the plates together with 100 μL of abscisic acid *d*_6_, which was used as an internal standard. Sample clean-up was performed with 500 μL of 1.5 M formic acid and 0.5% methanol and elution was achieved using 1 mL of methanol acidified with 0.1% of formic acid. The eluted fraction was then evaporated to dryness under a nitrogen stream at room temperature, and the evaporate was reconstituted with 100 μL of 0.05% formic acid in water. This was then vortexed for 20 min and filtered prior to injection into the chromatograph.

Calibration curves were prepared following the same procedure by spiking synthetic urine with increasing concentrations of the mixture of phenolic standards.

Quantitation of MPMs was achieved by means of separation by liquid chromatography with a Kinetex F5 100Å (50 × 4.6 mm i.d., 2.6 µm) (Phenomenex, Torrance, CA, USA) and detection with high-resolution mass spectrometry with an LTQ Orbitrap Velos mass spectrometer (Thermo Scientific, Hemel Hempstead, UK) equipped with an ESI source working in negative mode, as explained elsewhere [11]. Five µL of sample were injected into the reverse phase chromatographic column, which was kept at 40 °C, and gradient elution was performed with two phases (A: water (0.05% formic acid) and B: acetonitrile (0.05% formic acid)) at a constant flow rate of 0.5 mL/min using the following non-linear gradient: 0 min, 2% B; 1 min, 2% B; 2.5 min, 8% B; 7 min, 20% B; 9 min, 30% B; 11 min, 50% B; 12 min, 70% B; 15 min, 100% B; 16 min, 100% B; 16.5 min, 2% B; 21.5 min, 2% B. Total run time was 21.5 min.

Mass spectra were acquired in profile mode with a setting of 30,000 resolution at *m*/*z* 400, and the mass range was from *m*/*z* 100 to 2000. Operation parameters were as follows: source voltage, 5 kV; sheath gas, 50 units; auxiliary gas, 20 units; sweep gas, 2 units; and capillary temperature, 375 °C. Data acquisition was performed by Fourier transformed mass spectrometry (FTMS) mode (scan range *m*/*z* 100–1000) in combination with product ion scan experiments (MS^2^) (resolution range 15,000–30,000 FWHM).

MPMs were identified by comparison with the retention time of the standards in the case of the aglycones or by comparison with MS/MS spectra found in the literature for the accurate mass with an error of 5 ppm when aiming for the phase II metabolites. Trace Finder software version 4.1 (Thermo Fisher Scientific, San Jose, CA, USA) was used for chromatographic analysis. Due to the unavailability of phase II metabolite standards, the phase II metabolites were quantified in their respective aglycone equivalents. Samples that had MPM concentrations over the highest point of the calibration curve were diluted and reanalyzed. Due to the high variability of MPM identified in participants, probably derived from metabolic and metabotype differences, only metabolites that had less than 20% of values below the limit of quantitation were considered for statistical analyses. Values below the limit of detection were replaced by half of the limit of detection, and values below the limit of quantitation were replaced by the midpoint between the limit of detection and the limit of quantitation.

#### 2.3.3. Creatinine Analysis

Creatinine was measured by an adapted Jaffé alkaline picrate method for 96-well plates [12], because creatinine concentrations of spot urine samples can be used to assess urinary excretion of compounds in the absence of disease [13,14]. MPM concentrations were expressed as µg MPM/g creatinine.

### 2.4. Covariates and Other Variables

Sociodemographic and lifestyle habits such as physical activity, individual and family medical history, medical conditions, and medication use were assessed with self-reported questionnaires supervised by trained personnel [7]. Anthropometric and blood pressure measurements were also taken by trained staff. Energy intake in kcal/day was estimated from a validated food frequency questionnaire [8] and the dietary quality was rated based on the 14-item Mediterranean diet adherence score [15].

### 2.5. Statistical Analyses

MPM concentrations were transformed with a rank-based inverse normal method to approximate a normal distribution [16], and their one-year changes (difference between one-year and baseline) were analyzed as both continuous variables [1-standard deviation (1-SD)] increment and tertiles. Baseline characteristics per case status are described as means and standard deviations for continuous variables and percentages for categorical variables. Significant differences between controls and cases were calculated by analysis of t-test for continuous variables and the χ^2^-test for categorical variables.

To examine the association between changes of MPMs and type 2 diabetes, multivariable logistic regression models were conducted to estimate odds ratios (ORs) and their 95% confidence intervals. All analyses were adjusted in an increasing complexity manner for sex (women/men), age (years) and intervention group (MeDiet + olive oil, MeDiet + nuts, or control) (basic model), body mass index (kg/m^2^), physical activity (metabolic equivalent tasks in min/day), smoking status (current, former, or never), education level (primary education or secondary/higher education), baseline hypertension (yes/no), dyslipidemia (yes/no), and energy intake (kcal/day) (model 1) and baseline fasting plasma glucose (<100 mg/dL, 100–125 mg/dL, or >126 mg/dL) (model 2). We used robust variance estimators to account for recruitment center variability in all models.

The Simes procedure was used to account for multiple testing by adjusting the *p* values of the multivariable-adjusted associations between one-year changes and type 2 diabetes risk [17]. Analyses were performed with Stata software, version 16.0 (Stata Corp LP, College Station, TX, USA). Significance testing was considered for *p* < 0.05.

## 3. Results

The general characteristics of the study population at baseline (*n* = 172) are summarized in Table 1 according to whether they developed (cases) or not (controls) type 2 diabetes during a mean 3.6-year follow-up. According to the design of the cohort, most participants were dyslipidemic and hypertensive and were overweight or obese. Even though fasting plasma glucose at baseline was significantly higher in cases compared to the controls, none of the participants were receiving hypoglycemic agents.

The associations between individual urine MPMs and the odds of developing type 2 diabetes are presented in Table 2. Compared to participants at the lowest tertile of the one-year to baseline difference of hydroxybenzoic acid glucuronide, those in the highest tertile had significantly decreased odds of developing type 2 diabetes (OR [95% CI], 0.39 [0.23–0.64]; *p* < 0.001 for trend). A 1-SD increment in the one-year change of this same MPM concentration was also associated with reduced odds of developing type 2 diabetes. Although the trend was not maintained for the highest tertile, participants from the middle tertile of 4-hydroxybenzoic acid, hydroxytyrosol sulphate, and vanillic acid sulphate concentration changes had significantly diminished odds of developing type 2 diabetes compared to those in the lowest tertile. All these associations remained statistically significant after adjusting for multiple comparisons (Figure 2).

Because of the differences in baseline plasma glucose, which is a predictor of type 2 diabetes, between the cases and controls, we created an extra model further adjusted for fasting plasma glucose. After this adjustment, most of the differences were not maintained. Only participants from the middle tertile of 4-hydroxybenzoic acid concentration changes stayed at significantly decreased odds of developing type 2 diabetes compared to those in the lowest tertile. In addition, participants from the middle tertile of *m*-coumaric acid concentration changes showed to be at significantly increased odds of developing type 2 diabetes compared to the participants in the lowest tertile.

## 4. Discussion

In the present study we have described how changes in urinary MPMs were associated with the likelihood of developing type 2 diabetes in an older population at high risk of CVD after 3.6 years of follow-up. Our findings suggest that the changes in MPM concentrations, such as hydroxybenzoic acid glucuronide, found in urine might be early biomarkers of type 2 diabetes development or that these molecules may play a role in the prevention of the disease.

The association of polyphenol intake with various health outcomes has been studied for some time, and benefits have usually been suggested. MPMs, which are the products of the metabolism of polyphenols and the fingerprint of what has actually been circulating in the body, have been much less investigated. Recently, these molecules have been variably associated with type 2 diabetes [18,19,20,21]. To our knowledge, ours is the first study conducted within a nutritional intervention trial in which MPMs have been quantified in urine samples of an older population at high risk of developing cardiometabolic diseases. Other studies have investigated the association of MPMs with type 2 diabetes in different populations, with mixed results [18,19,20,21]. In addition, assessing one-year changes allows for interpreting the results at a longitudinal level. After adjusting for age, sex, intervention group, body mass index, physical activity, smoking status, education level, hypertension, dyslipidemia, and energy intake, higher one-year changes in the concentration of hydroxybenzoic acid glucuronide were associated with lowered odds of developing type 2 diabetes. Hydroxybenzoic acid is a final catabolic derivative from flavonoid and non-flavonoid metabolism. The production of this molecule has been attributed to the human colonic microbiota after in vitro studies with human fecal microbiota [22] and evaluation of microbial extracts with antiproliferative activity [23]. Phase II metabolites of hydroxybenzoic acid, such as its glucuronides, have been detected in urine after consumption of orange juice [24] and red raspberries [25], or after an organic diet [26]. 4-hydroxybenzoic acid, one of the modified but non-conjugated forms, has also been described as a derivative of pelargonidin thanks to studies with oral gavage of that particular polyphenol in animal models [27] and of cyanidin-3-glucoside after oral administration of this other polyphenol in clinical studies [28].

The interest of hydroxybenzoic acid is that one of its isomers, 4-hydroxybenzoic acid, is a precursor of ubiquinone, also known as Coenzyme Q, a key element of the mitochondrial electron transport chain [29]. The deficiency of this coenzyme has been suggested to be related to impairment of glucose tolerance [30,31]. A number of studies have assessed the association between ubiquinone concentrations and different measurements of diabetes or insulin resistance and have suggested that it as an important factor in maintaining glucose homeostasis. In a randomized, controlled trial of 78 prediabetic adults, a population similar to ours, the insulin resistance index estimated by the homeostasis model assessment (HOMA-IR), an important diabetes marker, decreased significantly in patients who received 200 mg/day of ubiquinone for eight weeks [32]. In this group, the free oxygen radical levels decreased significantly as well. Reactive oxygen species have historically been related to the development of diabetes and other non-communicable or inflammatory diseases [33]. Another randomized, controlled trial with 31 well-trained college athletes tested the supplementation of 300 mg/day of ubiquinone for 12 weeks. After the intervention, the investigators found similar results: lower HOMA-IR levels together with lower glycated hemoglobin and higher quantitative insulin sensitivity check index (QUICKI) in the group with higher white blood cell ubiquinone levels [34]. Other studies have described a reduction of glycated hemoglobin [35,36] and also negative correlation with insulin and HOMA-IR, and positive with QUICKI index, [36] after supplementation of 100 to 200 mg/day of ubiquinone to patients with type 2 diabetes. In the last study, increased catalase and glutathione peroxidase activities, enzymes that oversee detoxification of reactive oxygen species from the body, were found after ubiquinone supplementation. In other studies, ubiquinone has been proposed to improve insulin sensitivity by modulating the insulin receptor and glucose transporter GLUT4 [37]. The evidence from these studies suggests a potential role of hydroxybenzoic acid in the control of glucose homeostasis and improvement of diabetic control.

An increase in 4-hydroxibenzoic acid was observed in urine after an intervention with organic products (more than 80% of the diet coming from organic supermarkets) carried out by our group [26], as well as an increase of constituents from the hydroxybenzoic acid family after different types of (polyphenol-rich) foods such as oranges, cocoa, and almonds, or beverages like green tea, coffee, and red wine [6]. In addition, a prior study within the PREDIMED cohort showed that intake of hydroxybenzoic acid, among other (poly)phenols, was correlated with a lower incidence of type 2 diabetes [38]. These results lead to the hypothesis that not only might this polyphenol family be related to type 2 diabetes, but its circulating concentrations are related to the dietary pattern and the gut microbial population [39].

Changes in both hydroxytyrosol sulphate and vanillic acid sulphate were also identified as having an impact on type 2 diabetes onset, at least when the increase was small, between the first and second tertiles in model 1. Unlike with hydroxybenzoic acid, the evidence on these polyphenol species from human studies is rather scarce. Nevertheless, in 2011, the European Food Safety Authority (EFSA) approved the use of 5 mg/day of hydroxytyrosol or its derivatives for reduction of the risk of oxidative damage, among other effects [40].

Coming from a large cohort, the experimental analyses reduced the sample size, which represents the main limitation of this study. In addition, the variability given by the possible different metabotypes of the microbial populations could not be accounted for and there is a possibility of selection bias due to the lack of baseline samples of many PREDIMED subjects in the selection process of this study. Nevertheless, the precise MPM extraction from human urine and the novel methodology for high precision identification and quantitation are strengths of this study. The longitudinal analysis is an added strength.

Although fasting plasma glucose remains a reliable predictor of type 2 diabetes development, as seen in the baseline characteristics of the population, the results from this study provide a valuable human perspective support for the hypotheses that MPMs are associated with type 2 diabetes. Hence, the use of foods rich in polyphenols as prebiotics that lead to the production of these compounds by the microbiota may be a good strategy for curtailing type 2 diabetes and its associated burden.

## 5. Conclusions

In summary, changes of concentration of MPMs in urine in a sub-sample of participants from the PREDIMED study after one year of intervention were associated with type 2 diabetes incidence. To confirm the validity of these results, studies with larger sample sizes are warranted. Mechanistic studies would also be needed to understand the molecular basis of the observed associations.

## Figures and Tables

**Figure 1 antioxidants-11-01540-f001:**
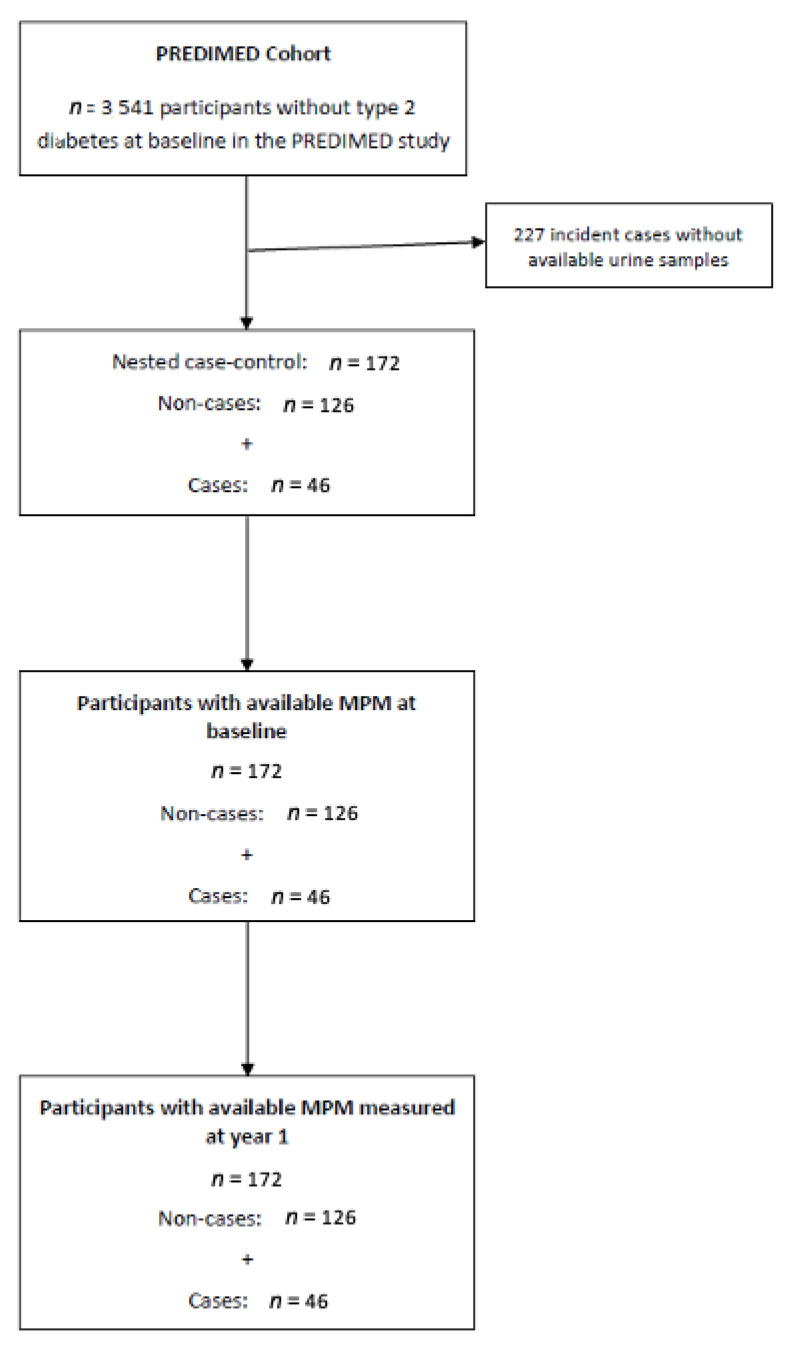
Flow-chart of the case-control design.

**Figure 2 antioxidants-11-01540-f002:**
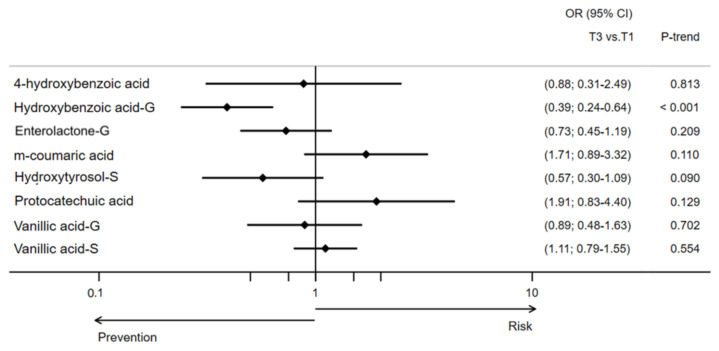
Likelihood (OR [95% CI]) of type 2 diabetes development by one-year changes in urinary concentrations of MPM in the PREDIMED Study. This model was adjusted for sex, age, intervention group, body mass index, physical activity, smoking status, education level, hypertension, dyslipidemia, and energy intake.

**Table 1 antioxidants-11-01540-t001:** General characteristics of the study population at baseline.

	Cases (*n* = 46)	Controls (*n* = 126)	*p*-Value
Women, *n* (%)	26 (56.5)	84 (66.8)	0.220
Age (years)	65.9 ± 6.0	67.9 ± 5.7	0.039
Intervention group, *n* (%)			0.334
Mediterranean Diet + EVOO	18 (39.1)	54 (42.9)	
Mediterranean Diet + nuts	12 (26.1)	42 (33.3)	
Control	16 (34.8)	30 (23.8)	
Dyslipidemia, *n* (%)	38 (82.6)	98 (77.8)	0.491
Hypertension, *n* (%)	45 (97.8)	113 (89.7)	0.084
BMI (kg/m^2^)	30.9 ± 3.0	30.6 ± 3.9	0.721
Energy intake (Kcal/day)	2381 ± 544	2276 ± 488	0.226
Smoking habit, *n* (%)			0.682
Current smoker	8 (17.4)	20 (15.9)	
Past smoker	12 (26.1)	26 (20.6)	
Never smoker	26 (56.5)	80 (63.5)	
Physical activity (METs-min/day)	258.2 ± 176.3	218.4 ± 208.6	0.250
Level of education, *n* (%)			
High and medium studies	12 (26.1)	25 (15.8)	0.378
Fasting plasma glucose (mg/dL)	118.8 ± 18.1	96.5 ± 12.8	<0.001

EVOO, extra-virgin olive oil; MET, metabolic task equivalents. Values are percentages for categorical variables and means ± SD for continuous variables. *p*-values were calculated by *t*-test for continuous variables and the χ^2^-test for categorical variables.

**Table 2 antioxidants-11-01540-t002:** Likelihood (OR [95% CI]) of incident type 2 diabetes by tertiles of one-year changes in urinary concentrations of MPMs in the PREDIMED Study.

No. of Cases		T1	T2	T3	*P* _Trend_	1-SD Increment
		58	57	57		
4-Hydroxybenzoic acid	Basic model	1.00 (ref)	0.31 [0.18–0.54]	0.81 [0.38–1.71]	0.579	0.91 [0.70–1.19]
	Multivariable model 1	1.00 (ref)	0.31 [0.19–0.53]	0.88 [0.31–2.49]	0.813	0.94 [0.70–1.26]
	Multivariable model 2	1.00 (ref)	0.22 [0.14–0.36]	0.67 [0.30–1.48]	0.284	0.78 [0.60–1.01]
Hydroxybenzoic acid glucuronide	Basic model	1.00 (ref)	0.54 [0.34–0.86]	0.41 [0.26–0.65]	<0.001	0.63 [0.51–0.79]
	Multivariable model 1	1.00 (ref)	0.52 [0.32–0.86]	0.39 [0.24–0.64]	<0.001	0.61 [0.49–0.77]
	Multivariable model 2	1.00 (ref)	0.49 [0.13–1.84]	0.61 [0.19–1.97]	0.341	0.69 [0.44–1.09]
Enterolactone glucuronide	Basic model	1.00 (ref)	0.97 [0.40–2.35]	0.79 [0.54–1.14]	0.208	1.08 [0.93–1.25]
	Multivariable model 1	1.00 (ref)	1.05 [0.51–2.13]	0.73 [0.45–1.19]	0.209	1.10 [0.89–1.36]
	Multivariable model 2	1.00 (ref)	1.17 [0.79–1.73]	0.79 [0.29–2.13]	0.789	1.06 [0.73–1.54]
m-coumaric acid	Basic model	1.00 (ref)	1.35 [0.75–2.44]	1.78 [0.82–3.87]	0.147	1.26 [0.97–1.63]
	Multivariable model 1	1.00 (ref)	1.48 [0.75–2.91]	1.71 [0.89–3.32]	0.110	1.24 [0.99–1.55]
	Multivariable model 2	1.00 (ref)	1.46 [1.13–1.89]	1.58 [0.91–2.73]	0.110	1.11 [0.96–1.30]
Hydroxytyrosol sulphate	Basic model	1.00 (ref)	0.54 [0.41–0.72]	0.57 [0.22–1.50]	0.258	0.93 [0.58–1.49]
	Multivariable model 1	1.00 (ref)	0.59 [0.51–0.68]	0.57 [0.30–1.09]	0.090	0.94 [0.63–1.39]
	Multivariable model 2	1.00 (ref)	0.38 [0.17–0.86]	0.34 [0.07–1.72]	0.265	0.71 [0.31–1.65]
Protocatechuic acid	Basic model	1.00 (ref)	1.51 [0.40–5.67]	1.75 [0.73–4.20]	0.208	1.26 [0.87–1.83]
	Multivariable model 1	1.00 (ref)	1.83 [0.54–6.15]	1.91 [0.83–4.40]	0.129	1.30 [0.91–1.87]
	Multivariable model 2	1.00 (ref)	3.00 [0.75–12.02]	2.08 [0.89–4.88]	0.068	1.35 [0.85–2.13]
Vanillic acid glucuronide	Basic model	1.00 (ref)	0.99 [0.48–2.04]	0.76 [0.31–1.84]	0.546	0.94 [0.60–1.48]
	Multivariable model 1	1.00 (ref)	0.95 [0.39–2.35]	0.89 [0.48–1.63]	0.702	1.04 [0.71–1.50]
	Multivariable model 2	1.00 (ref)	1.17 [0.44–3.10]	1.53 [0.61–3.81]	0.399	1.31 [0.82–2.08]
Vanillic acid sulphate	Basic model	1.00 (ref)	0.83 [0.74–0.93]	1.13 [0.71–1.81]	0.608	1.02 [0.75–1.38]
	Multivariable model 1	1.00 (ref)	0.87 [0.77–0.97]	1.11 [0.79–1.55]	0.554	1.01 [0.79–1.31]
	Multivariable model 2	1.00 (ref)	0.76 [0.29–2.01]	1.38 [0.97–1.97]	0.046	1.06 [0.70–1.61]

Inverse normal transformation was applied to raw values of metabolites. The basic model was adjusted for sex, age, and intervention group. To the covariables in the basic model we added body mass index, physical activity, smoking status, education level, hypertension, dyslipidemia, and energy intake to build model 1. Model 2 was further adjusted for baseline fasting plasma glucose. Robust variance estimators were used to account for recruitment center. CI indicates confidence interval; MPM, microbial phenolic metabolite; SD, standard deviation; OR, odds ratio; Ref., reference. In bold, tertiles that were significantly different from the reference after adjusting the *p* values to account for multiple testing using the Simes procedure.

## Data Availability

There are restrictions on the availability of data for the PREDIMED trial due to the signed consent agreements around data sharing, which only allow access to external researchers for studies following project purposes. Requestors wishing to access the PREDIMED-Plus trial data used in this study can make a request to the PREDIMED trial Steering Committee chair: restruch@clinic.cat. The request will then be passed to members of the PREDIMED Steering Committee for deliberation.

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
