# Peer review of "One-Year Changes in Urinary Microbial Phenolic Metabolites and the Risk of Type 2 Diabetes—A Case-Control Study"

_antioxidants, 2022, doi:10.3390/antiox11081540_

Round 1
Reviewer 1 Report
This manuscript seeks to associate the concentration of various gut microbial phenolic metabolites with risk of developing type 2 diabetes. The researchers use a subset of randomly selected data from the robust PREDIMED study.
Overall, this manuscript has well-designed methods and is well-written. It will generate interest within the research community because of it's identification of a potential biomarker for future type 2 diabetes development. I have only a few minor suggestions for the authors to consider in revising their manuscript.
1. Why were the specific MPMs chosen in this study? There are hundreds (if not thousands) of phenolic metabolites generated by the gut microbiota, not to mention the plethora of phase 2 metabolites subsequently formed. Without untargeted metabolomics and massive computing power, it is impossible to evaluate the potential association of every possible metabolite with outcomes, so it is very reasonable to select only a handful to monitor. I am just curious as to how these particular metabolites were selected?
2. It appears the urine samples were collected at two timepoints (baseline and 1 year later). Do all the data represent changes in MPMs over that 1 year time frame? Or is it just the concentration of MPMs at the endpoint? Additionally, the authors mention that subjects were followed for ~3.6y; I assume this is for diabetes development? If so, please clarify in the text so the reader understands the difference between the 1 year time frame of the urine collections and the ~3.6y follow up for diabetes development.
3. In section 2.3.1, were urine samples collected under fasting conditions? Were they all collected from participants consuming the same diet the day before? Also, were urine samples acidified prior to freezing to preserve phenolic metabolites?
4. Line 160, what does "[articulo emy]" mean?
5. While the difference in risk between the tertiles is interesting, I'm curious what the actual magnitude of difference was between these tertiles? If there was a large quantitative difference in the amount of hydroxybenzoic acid glucuronide between tertiles 1 and 3, then that makes it a promising biomarker. However, if the quantitative difference is small, that makes it more difficult to use in clinical practice, despite the statistical significance seen here...
6. Line 292 - define "organic diet".
Author Response
Thank you very much for the remarks.
- When we developed the targeted method, we had to decide, as you point out, between a massive amount of molecules. We tried to choose a set that would represent more or less all groups.
- As explained in section 2.5 the data represent the differences between 1-year and baseline. In section 3 line 223 we explain that the mean follow up for diabetes development was 3.6y.
- We have added that the urines were taken in fasting conditions. They were not acidified prior to snap freezing but they were right after thawing. The diets prior sampling were collected by trained dietitians but for reference purposes, not for analysis.
- This was a typo, a reference has been added.
- Because the data ara SD-transformed the raw difference is not informative. We show in the attached file the predicted differences for likelihood of type 2 diabetes development by using the margins function and the differences of the estimations between groups by using the contrast function. Both in model 2.
- We have added the definition for organic diet in that paper in line 324.

Reviewer 2 Report
Anti-oxidants-1828966
One -year changes in urinary microbial phenolic metabolites (MPM) and the risk of type 2 diabetes. A case-control study
Maria Marhuenda-Munoz, et al.
Overall: This is a case-control study using a subset of the data and samples of the PREDIMED trial. The novel twist to this kind of typical study is the use of a subset population of a large clinical trial to quantify the MPMs in urine and assessing their association with type 2 diabetes risk. Interesting findings were seen in 1-year changes in hydroxybenzoic acid glucuronide in comparing the lowest tertile to the highest tertile in the probability of developing type 2 diabetes. The caveat is that when data were adjusted for fasting plasma glucose, the statistical significance was lost.
Specific Comments:
1.This is an interesting study in that most data in the literature deal with the type of polyphenol in the diet that might affect some biological conditions such as development of cardiovascular disease, cancer diabetes etc. Rarely are the microbial flora of these polyphenols examined for their contribution to changes in the probability of developing these diseases.
2. The research group conducting this study is outstanding and well published. They have made some seminal observations of the Mediterranean diet and development of heart disease.
3. As pointed out by the authors, the small sample size represents the main limitation of the study in terms of showing statistical significance of the findings, when using fasting glucose levels
4. Nonthe-less, I agree with the authors that the MPM extractions from human urine and the longitudinal analysis are strong points of the study
5. I would hope that there is follow-up in this population in quantification of the microbial population during the time of dietary changes and linking this to the MPMs and development of type2 diabetes. And yes a larger population size is definitely need for these studies.
Author Response
Thank you for your kind comments on our article. We hope to continue studying the effect of MPM on human health in the following years in order to reduce the use of more invasive sampling. We are working on reducing the length of the analyses in order to be able to increase the number of samples that can be analyzed in one batch. In addition, we are interested in studying MPM in younger populations to see their effect prior to the increase in fasting plasma
glucose levels. Again, thank you very much for the review.
Reviewer 3 Report
This is a well written paper. No major concerns
MINOR CONCERNS
1) The references are not listed in a consistent style. Some titles are in lower case; others are capitalized. When titles are capitalized they are often capitalized incorrectly. Suggestion: Cite all titles in lower case. It's easier.
Reference 16: "Gunnar" should probably be abbreviated to "G" to be consistent.
Table 1, BMI (kg/m2) "2" should be in superscript.
Author Response
Thank you for the remarks. We have rechecked the references and corrected Table 1.